# Long-Term Effects of Ochratoxin A on the Glutathione Redox System and Its Regulation in Chicken

**DOI:** 10.3390/antiox8060178

**Published:** 2019-06-17

**Authors:** Benjámin Kövesi, Mátyás Cserháti, Márta Erdélyi, Erika Zándoki, Miklós Mézes, Krisztián Balogh

**Affiliations:** 1Department of Nutrition, Szent István University, H-2103 Gödöllő, Hungary; benjamin.kovesi@gmail.com (B.K.); Erdelyi.Marta@mkk.szie.hu (M.E.); Balogh.Krisztian@mkk.szie.hu (K.B.); 2Department of Environmental Safety and Ecotoxicology, Szent István University, H-2103 Gödöllő, Hungary; Cserhati.Matyas@mkk.szie.hu; 3MTA-KE-SZIE Mycotoxins in the Food Chain Research Group, Kaposvár University, H-7400 Kaposvár, Hungary; Zandoki.Erika@mkk.szie.hu

**Keywords:** Ochratoxin A, glutathione, glutathione-peroxidase, oxidative stress, gene expression

## Abstract

The purpose of this study was to evaluate the effect of three-weeks ochratoxin A (OTA) exposure on some lipid peroxidation parameters, reduced glutathione concentration and glutathione-peroxidase activity, as well as expression of oxidative stress response-related (*KEAP1*, *NRF2)* and glutathione system (*GPX3*, *GPX4*, *GSS*, *GSR)* genes in chickens. Three levels of exposure (106, 654 and 1126 μg/kg feed) were applied. The results showed that OTA initiated free radical formation, which was suggested by the increase in the malondialdehyde content in the liver and kidney, which was more marked in the liver, depending on the length of exposure and dose. Reduced glutathione concentration increased as an effect of the highest OTA dose in blood plasma and in liver, but not in red blood cell hemolysates and the kidney. Glutathione peroxidase activity did not change in the blood and showed increasing tendency in the liver, and significant increase in the kidney. Expression of *KEAP1* gene showed up-regulation in the liver, and down-regulation in the kidney, but overexpression of *NRF2* gene was found in the liver and kidney at the highest dose. However, down-regulation of Nrf2 dependent genes, *GPX3*, *GPX4*, *GSS* and *GSR*, suggested an improper antioxidant response at the protein level, thus oxidative stress occurred, even at the dose of the EU regulatory limit for poultry diets.

## 1. Introduction

Ochratoxin A (OTA) is a secondary metabolite of certain species of the *Aspergillus* and *Penicillium* genus [1]. The chemical structure of OTA consists of weak organic acids with a dihydroisocumarin moiety joined by a peptide bond to 1-phenylalanine [2]. There are three ochratoxin forms, designated as A, B and C, which have slight structural differences; however, ochratoxin A (OTA) is chlorinated and is the most toxic one [3]. OTA contamination in Europe in the year 2018, according to the recent Biomin World Mycotoxin Survey Report [4] was 40% in the finished (complete) feeds and 12% in cereals, with the average of positive samples containing 4 μg/kg in finished feeds and 19 μg/kg in cereals, respectively. OTA has a similar structure to the amino acid phenylalanine, therefore it can impair protein synthesis [5]. OTA is very stable during feed-processing, hence it cannot be eliminated from the feed industry, and remains intact and biologically active in the finished feed. Chicken is sensitive to OTA, LD_50_ value at 21 days of age is 3.6 ± 0.6 mg/kg b.w. [6]. 

OTA has nephrotoxic [7,8], mutagenic [9], carcinogenic [9], teratogenic [10] and immunosuppressive [11,12] effects in animals and humans. However, not all of the aforementioned toxic effects have been proven in poultry species. An immune suppressive effect was found even at the European Union maximum proposed level (0.1 mg/kg feed) in broiler chicken [13]. After absorption, OTA was present at the highest amount in the blood, followed by—in a decreasing order—the kidneys, liver, muscle and adipose tissue [14]. Feeding OTA-contaminated feed to broiler chicken (2.5 mg OTA/kg) caused a significant reduction in body-weight gain, and the relative kidney weight increased [15]. 

OTA affects the expression of several genes related to cell damage, apoptosis, cellular stress and antioxidant defence systems [16]. The mechanism of OTA toxicity includes the formation of oxygen free radicals and consequently peroxidation of polyunsaturated fatty acids [17]. This is supported by those results which are proving that low molecular weight antioxidants, such as tocopherol or ascorbic acid, decrease the formation of lipid peroxides and the corresponding toxic effects of ochratoxin A [18]. Molecular mechanisms responding to oxidative stress are conserved in vertebrates, including poultry. The master regulator of the oxidative stress response is the transcription factor nuclear factor-erythroid 2 p45-related factor 2 (Nrf2) [19] in connection with Kelch-like ECH-associated protein 1 (Keap1) [20]. OTA-induced oxidative stress has an effect on both enzymatic and non-enzymatic antioxidant defences [21], and also modulates Nrf2 gene and protein expression, which regulates the expression of the Antioxidant Response Element gene cluster [22]. In response to an elevated ROS level, Nrf2 induces the expression of antioxidant enzymes, which are the key component of the glutathione (GSH) biosynthesis pathway [23]. Changes in the redox balance of the cells regulate the protein expression of Nrf2 and its activity. Under unstressed conditions, Nrf2 is bounded to Keap1 to promote Nrf2 ubiquitination and further enzymatic degradation in proteasomes [24], resulting in low basal Nrf2 activity. However, in an oxidative stress situation the activity increases, due to the oxidation of Keap1 cysteine side chains, as an oxidative stress sensor. Consequently, the interaction between Nrf2 and Keap1 destabilizes and allows nuclear translocation of Nrf2 to transcribe its specific target genes [25]. Many Nrf2-regulated enzymes are involved in the antioxidant defence [26] and expression of most of them is inhibited by OTA treatment in the kidney of chicken [27]. However, there is no data available about the effect of OTA on glutathione redox system.

The purpose of the present study was to evaluate the toxic effect of OTA related to peroxidation of unsaturated fatty acids and on the other side, the oxidative stress response in the liver and kidney of chickens. For this purpose, a sub-chronic in vivo toxicological experiment was applied, in which markers of peroxidation of lipids, amount and activity of the glutathione redox system and expression of several genes regulating their biosynthesis or repair were analysed. 

## 2. Materials and Methods

### 2.1. Production and Determination of Ochratoxin A

Ochratoxin A was produced by artificial infection of sterile ground corn substrate with an *Aspegillus albertensis* strain (SZMC 22107) deposited in the Microbiological Collection of the University of Szeged. Ochratoxin A was determined by the high performance liquid chromatography (HPLC) method with fluorescence detection according to [28] after immune-affinity clean-up with OchraStar R IAC column (RomerLabs, Tulln) in triplicates. Predicted and measured OTA content of the complete feed is shown in Table 1.

### 2.2. Animals, Treatments and Sample Preparations

A total of 78 broiler chickens (21 days of age) were used for the experiment. Animals had free access to feed and water. Effect of OTA toxicity was investigated in three different dosage groups: Control; low (O1: 106 μg/kg feed), medium (O2: 654 μg/kg feed) and high (O3: 1126 μg/kg feed). The doses were calculated based on the European Commission recommendation (2006/576/EC) for OTA in chicken complete feed (100 μg/kg), namely 1×, 5× and 10× of the recommendation levels were used as predicted doses, which were validated by analysis (Table 1). An OTA-contaminated diet was fed for 21 days. At each sampling, on 28, 35 and 42 days of age, six animals from each experimental group were investigated. Birds were euthanized by cervical dislocation, blood was collected on ice, the liver and kidney were removed, collected on ice, and all samples stored at −70 °C until analysis. For gene expression studies, portions of liver and kidney were frozen in liquid nitrogen immediately after sampling, and stored at −70 °C until the analysis to prevent RNA degradation.

All procedures were conducted in accordance with the guidelines set by the European Communities Council Directive (86/609 EEC) and the protocol was approved by the Food Chain Safety, Land use, Plant and Soil Protection and Forestry Directorate of the Pest County Governmental Office (PE/EA/1964-7/2017) with the limitation to use as low a number of animals as possible for an accurate statistical analysis.

### 2.3. Determination of Lipid Peroxidation and Antioxidant Parameters

Markers of the initial phase of lipid peroxidation in the liver, the amount of conjugated dienes (CD) and conjugated trienes (CT), was measured by the absorbance of samples at 232 nm and 268 nm after extraction in 2,2,4-trimethylpentane (Reanal, Budapest) [29]. The other analyses were made from blood plasma, red blood cell hemolysate (1:9 *v*/*v*) in redistilled water, and in tissue homogenates which was prepared with 9-fold volume of cold (4 °C) physiological saline (0.65% *w*/*v* NaCl). Thiobarbituric reactive substances content, was used as the marker of terminal phase of lipid peroxidation and was measured in the native homogenate of liver and kidney according to the method of [30], and expressed as malondialdehyde, which served as the standard (1,1,3,3 tetraethoxypropane, Fluka, Buchs). Reduced glutathione (GSH) concentration and glutathione-peroxidase (GPx) activity was measured in the 10,000 g supernatant fraction of liver and kidney homogenates by Ellmann’s reagent (5,5’-dithiobis-(2-nitrobenzoic) acid, Sigma, St. Louis) according to the method of [31], and according to [32] using cumene-hydroperoxide (Sigma, St. Louis) and reduced glutathione (Reanal, Budapest) as co-substrates, respectively. GSH content and GPx activity were calculated to protein content, which was determined by the biuret method in blood plasma and red blood cell hemolysate [33], and using Folin-phenol reagent (Merck, Darmstadt) according to the method of [34] in the 10,000 g supernatant fraction of liver and kidney homogenates.

### 2.4. Quantitative Real-Time PCR

The purification of total RNA was carried out by NucleoZOL reagent (Macherey-Nagel, Düren) from 6–10 mg liver homogenates, according to the manufacturer’s instructions. DNase I treatment was performed after RNA extraction according to the protocol of the supplier (Thermo Fisher Scientific, San Jose) to remove any genomic DNA contamination. Agarose gel electrophoresis and NanoPhotometer (Implen, Munich) measurement was carried out to verify the quantity and integrity of total RNA samples. Furthermore, only those samples were accepted, which had the ratios of absorption 260:280 nm at >2.0. The cDNA production was implemented by RevertAID reverse transcriptase (Thermo Fisher Scientific, San Jose) and random nanomer primer from 1 µg of total RNA as described by the standard protocol. The primers used for the quantification of the mRNA transcriptional levels of the target (*KEAP1*; *NRF2*; *GPX3*: Extracellular glutathione peroxidase; *GPX4*: Phospholipid hydroperoxide glutathione peroxidase, *GSS*: Glutathione synthetase and *GSR*: Glutathione reductase) and endogenous control genes (*GAPDH*) are shown in Table 2 and were designed by Primer Express 3.0.1 (Thermo Fisher Scientific, San Jose). *GAPDH*, as an endogenous control gene, was chosen according to literature as it had no known interaction with oxidative stress and mycotoxin exposure in broiler chickens as it was applied as a housekeeping control gene in several other studies [35,36,37].

Quantitative real-time PCR measurements, in the case of *GPX4*, *GSS* and *GSR* genes, were performed in duplexes (*GAPDH* and one target gene), using MGB TaqMan probes shown in Table 3 (Thermo Fisher Scientific, San Jose) as described previously [38]. 

The measurements were carried out with pooled cDNA templates in both cases. The pools were formed from equal (100 ng) amounts of cDNA per 6 chicken livers of each sampled group at each sampling point of treatment in five technical replicates. Based on the results of previous experiments, no measurable differences were found if the determination was made from pooled and not individual samples. Moreover, no template controls (NTC) were applied for each of the PCR measurements. 

The PCR profile for the *GPX4*, *GSS*, *GSR*, *GPX3* and *KEAP1* target genes consisted of 95 °C for 10 min for pre-amplification denaturation (PAD), and 95 °C for 15 s, 58 °C for 30 s and 72 °C for 30 s for 45 cycles. The PCR profile for the *NRF2* target gene was 95 °C for 10 min for pre-amplification denaturation (PAD), and 95 °C for 15 s, 60 °C for 30 s and 72 °C for 30 s for 45 cycles. In case of the duplexes (*GPX4*, *GSS* and *GSR*) both VIC and FAM signals, while in the case of singleplexes (*GPX3*, *KEAP1* and *NRF2*) SYBR Green signals were detected at the end of the extension period. Amplified products were verified by gel electrophoresis, while in the case of singleplexes, melting curve analysis was also performed. The threshold cycle (Ct) of the target genes (*GPX4*, *GPX3*, *NRF2*, *KEAP1*, *GSS* and *GSR*) and the endogenous control gene *(GAPDH*) was determined by StepOne™/StepOnePlus™ Software v2.2 (Thermo Fisher Scientific, San Jose) applying the comparative Ct method. The delta Ct (ΔCt), delta-delta Ct (ΔΔCt) and relative quantification (RQ = 2^−ΔΔCt^) values were calculated by the formula described in [39].

### 2.5. Statistical Analysis

Data were expressed as means ± SD. The data were first subjected to a Kolmogorov–Smirnov normality test, and homogeneity of variance was confirmed by the Barlett test. Data passing both tests were analysed by one-way ANOVA followed by Tukey’s post hoc test. Statistical analysis was performed using GraphPad PRISM version 7 software (GraphPad, San Diego), and *p* ≤ 0.05 was considered significant.

## 3. Results

### 3.1. Clinical Findings

Clinical signs of toxicity and mortality were not observed during the trial in the experimental groups.

### 3.2. Effect of Long-Term Ochratoxin A Exposure on Lipid Peroxidation and Glutathione Redox System in Chicken Liver

Markers of the initial phase of lipid peroxidation, conjugated dienes and trienes, did not change significantly in the liver as an effect of different OTA doses (data not shown). The termination phase marker, a meta-stable end product (malondialdehyde), also did not change in blood plasma and red blood cell hemolysates (data not shown), but increased in the liver at all samplings, and at day 7 in the kidney of the highest (O3) dose group as compared to the control (Table 4). 

The non-enzymatic part of the glutathione redox system, GSH concentration, increased significantly as an effect of the highest dose of OTA exposure (O3) in blood plasma on day 7, and also in the 10,000 g supernatant fraction of liver homogenate on days 14 and 21 as compared to the control (Table 5). There were no significant alterations in GSH content of red blood cell hemolysates and kidney homogenates (data not shown). 

There were no differences in the GPx activity in blood plasma and red blood cell hemolysates (data not shown), while increasing tendency was found in liver homogenates at days 7 and 14 in the moderate and highest dose groups (O2 and O3), and a significant increase in kidney homogenates at day 7 in the highest (O3) dose group as compared to the control (Table 6). 

### 3.3. Effect of Long-Term Ochratoxin A Exposure on Expression of Genes Encoding Components of Glutathione System and Required for Their Synthesis or Repair in Liver and Kidney

The relative expression of the *KEAP1* gene was significantly higher in the liver on day 14 in the moderate and high dose groups (O2 and O3), but lower in kidney on days 7 and 14 in the O2 group, and on day 21 in the O1 and O2 groups than the control (Table 7). Overexpression of the *NRF2* gene was found in liver and kidney at all sampling times in the O3 group as compared to the control (Table 7). 

In the case of *GPX3* expression, opposite tendencies were found at day 7 between the liver and kidney. It was downregulated in the liver but upregulated in the kidney in the O1 group (Table 8). At day 14, similar changes were found in the liver and kidney, overexpression of *GPX3* was found in the O3 group. One week later, at day 21, *GPX3* expression was downregulated in all treatment groups in the liver, and upregulated in the kidney in the O3 group (Table 8). *GPX4* expression did not change markedly in the liver, but it was significantly downregulated at day 7 in O2 and O3 groups, and in all OTA treated groups at day 14 and 21 in the kidney. *GSS* gene expression was lower as an effect of the lowest (O1) dose in the liver at day 14. In the kidney, downregulation of the *GSS* gene was found on day 7 in the two higher dose groups (O2 and O3), and on day 14 in the lowest dose group (O1) as compared to the control (Table 8).

Expression of *GSS* and *GSR* genes were downregulated in the lowest dose group (O1) on day 14 in both the liver and kidney (Table 9). 

## 4. Discussion

The present in vivo sub-chronic experiment demonstrated that OTA exposition significantly increased free radical formation in the liver and kidney. OTA distributed into different tissues more or less equally after oral intake [39], therefore its effects were possibly the same at the same time in both tissues. Oxidative stress is proposed as the effect of OTA toxicity both in the liver [40] and kidney [41], and the oxidative reactions activate lipid peroxidation [42]. These findings were partly supported by the results of the present study because the initial phase of lipid peroxidation, amount of CD and CT, did not increase. However, it reached the termination phase, as proven by the higher MDA content, possibly due to the long-term period of exposure. The changes were more marked in the liver than in the kidney, but in both tissues only the highest OTA dose (1126 μg/kg feed) revealed significant changes. These results suggested that the cellular defence mechanism was unable to inhibit the lipid peroxidation processes at that dose. 

These results can be explained with the changes of the glutathione redox parameters, because neither GSH content nor GPx activity increased systematically during the period of OTA exposure. This meant that the amount or activity of the glutathione redox system was not adequate for the elimination of oxygen free radicals, in particular at high OTA exposure. 

On the other hand, the expression of the redox sensitive gene, *KEAP1*, was overexpressed in the liver and downregulated in the kidney. This meant that release of Nrf2, as a transcription activator of the Antioxidant Response Element (ARE) gene cluster [43], might have been released from the binding with Keap1 in the kidney, but much less efficiently in the liver. However, at the same time the relative expression of the *NRF2* gene increased in both the liver and kidney, as a response to oxidative stress, but hypothetically Nrf2 remains bounded to Keap1 in the liver, if the same changes occurred also at the protein expression level, however, it was not determined in this study. 

A higher level of unbound Nrf2 activates the ARE containing genes encoding glutathione metabolism enzymes, such as *GSS*, *GSR* and glutathione peroxidases (in this study *GPX3* and *GPX4*). Non-significant changes in GSH concentration together with downregulation of the gene expression of *GSS* suggests that the cellular concentration of GSH did not change, because GSS protein is required for the synthesis of GSH. This result was supported by previous studies where OTA exposure down-regulated the genes involved in GSH metabolism [9,21], but contradictory with another in rats [44], where a significant decrease of GSH concentration was found in the kidney. The possible cause of this contradiction would be the higher OTA exposure applied in the previous study, and probably another mechanism, for instance, the conjugation of OTA with GSH [45] which may also result in a lower concentration. *GSR* expression also did not increase as an effect of OTA exposure, which meant that the reduction of glutathione-disulphide (GSSG) to GSH was also not effective, which also supported the insufficient antioxidant response in the liver and kidney, therefore induction of oxidative stress and consequently lipid peroxidation. However, it should be noted that GSH homeostasis in the cell is regulated by its synthesis and/or recycling, and also by the rate of utilization and efflux [46]. *GPX3* encodes the extracellular glutathione peroxidase (GPx3), which mainly originates from kidney tubular cells [47]. The overexpression in the kidney proves the importance of the kidney as a source of GPx3, but an opposite tendency, downregulation, occurred in the liver, which has less importance in the synthesis of this GPx isoenzyme. This difference in the gene expression also suggests that regulation of glutathione peroxidase genes, in particular *GPX3*, in the kidney is more sensitive to oxidative stress caused by OTA exposure than in the liver. However, higher relative gene expression did not manifest at the protein level because there were no significant changes in GPx activity in the blood, even at overexpression of kidney-origin *GPX3,* which meant that such an increase did not cause a significantly higher amount of enzymatically active proteins. The other glutathione peroxidase enzyme, *GPX4,* expression at both gene and protein levels occurred in most of the cell types, due to its effect on the inhibition of membrane phospholipid oxidation [48]. However, its gene expression did not change in the liver, and was downregulated in the kidney. At the protein level, the activity of GPx did not change as an effect of OTA, which suggested that even in the case of downregulation of gene expression the enzyme activity remains stable, possibly due to the post-translation modification of the preformed enzyme proteins [49].

In conclusion, the results of the present study have revealed that OTA initiates free radical formation both in the liver and kidney, but not in blood. The results showed that lipid peroxidation depended on the length of exposure and the dose applied. Lack of ARE activation, which was suggested by the low mRNA level of most of the Nrf2 dependent genes, resulted in improper antioxidant defence, thus mild oxidative stress, even at the regulatory dose proposed by the European Commission for poultry diets. Based on the results, defining lower values for poultry diets can be proposed and attention should be placed on the use of feed additives, such as phytobiotics, which are useful for activation of the antioxidant gene cluster [50], possibly even in the case of OTA exposure.

## Figures and Tables

**Table 1 antioxidants-08-00178-t001:** Predicted and measured ochratoxin A concentration in the complete feed.

Group	Predicted OTA (μg/kg)	Measured OTA (μg/kg)
Control	0	<57
O1	100	106 ± 10.5
O2	500	654 ± 65.2
O3	1000	1126 ± 93.7

**Table 2 antioxidants-08-00178-t002:** Primers of target (*GPX4*, *GPX3*, *GSS*, *GSR*, *NRF2*, *KEAP1*) and endogenous control (*GAPDH*) genes.

Gene	Forward (5’-3’)	Reverse (5’-3’)	GenBank Accession Nr.
*GAPDH*	TGACCTGCCGTCTGGAGAAA	TGTGTATCCTAGGATGCCCTTCAG	NM_204305.1
*KEAP1*	CATCGGCATCGCCAACTT	TGAAGAACTCCTCCTGCTTGGA	XM_025145847.1
*NRF2*	TTTTCGCAGAGCACAGATAC	GGAGAAGCCTCATTGTCATC	NM_205117.1
*GPX3*	ATCCCCTTCCGAAAGTACGC	GACGACAAGTCCATAGGGCC	NM_001163232.2
*GPX4*	AGTGCCATCAAGTGGAACTTCAC	TTCAAGGCAGGCCGTCAT	NM_001346448.1
*GSS*	GTACTCACTGGATGTGGGTGAAGA	CGGCTCGATCTTGTCCATCAG	XM_425692.6
*GSR*	CCACCAGAAAGGGGATCTACG	ACAGAGATGGCTTCATCTTCAGTG	XM_015276627.2

**Table 3 antioxidants-08-00178-t003:** Dual labelled probes for target and endogenous control genes. MGB-NFQ quencher was used in all probes.

Gene	MGM-NFQ Dual Labelled Probe 5’-3’	Fluorescent Dye
*GAPDH*	CCAGCCAAGTATGATGAT	VIC
*GPX4*	CAGCCCAATGGAG	FAM
*GSS*	AGGAGGGAACAACCTG	FAM
*GSR*	CTGGACTTCGGCTC	FAM

**Table 4 antioxidants-08-00178-t004:** Effect of Ochratoxin A on malondialdehyde content in liver and kidney homogenates (mean ± SD; *n* = 6).

**Malondialdehyde in Liver Homogenate (μmol/g Wet Weight Tissue)**				
	Day 0	Day 7	Day 14	Day 21
Control	64.46 ± 11.23	49.89 ± 6.49 ^a^	43.71 ± 14.24 ^a^	58.21 ± 5.89 ^a^
O1		57.21 ± 11.54 ^ab^	59.83 ± 7.31 ^ab^	61.54 ± 14.95 ^ab^
O2		56.79 ± 13.57 ^ab^	62.34 ± 15.11 ^ab^	56.53 ± 9.91 ^a^
O3		77.59 ± 28.07 ^b^	74.28 ± 18.64 ^b^	77.41 ± 8.39 ^b^
**Malondialdehyde in Kidney Homogenate (μmol/g Wet Weight Tissue)**				
	Day 0	Day 7	Day 14	Day 21
Control	36.32 ± 10.10	41.11 ± 7.98 ^a^	41.03 ± 12.27	46.19 ± 5.37
O1		42.38 ± 9.19 ^ab^	42.55 ± 12.39	44.14 ± 2.67
O2		46.25 ± 11.94 ^ab^	43.65 ± 9.19	41.32 ± 9.68
O3		57.93 ± 7.44 ^b^	50.75 ± 8.14	45.23 ± 3.08

^a,b^ Different superscripts in the same column mean significant difference at *p* < 0.05 level. O1: 106 μg OTA/kg feed, O2: 654 μg OTA/kg feed, O3: 1126 μg OTA/kg feed.

**Table 5 antioxidants-08-00178-t005:** Effect of Ochratoxin A treatment on reduced glutathione concentration in blood plasma and liver homogenates (mean ± SD; *n* = 6).

**Blood Plasma (μmol/g Protein)**				
	Day 0	Day 7	Day 14	Day 21
Control	11.47 ± 1.91	11.70 ± 1.65 ^a^	11.91 ± 2.67	11.68 ± 1.00
O1		12.15 ± 2.42 ^a^	12.10 ± 1.74	12.42 ± 2.16
O2		14.51 ± 1.89 ^a^	13.28 ± 1.58	11.80 ± 1.50
O3		17.78 ± 1.29 ^b^	12.75 ± 2.00	13.26 ± 1.60
**Liver (μmol/g 10,000 g Supernatant Protein)**				
	Day 0	Day 7	Day 14	Day 21
Control	3.25 ± 0.37	5.22 ± 1.30	5.24 ± 0.53 ^a^	4.08 ± 0.53 ^ab^
O1		5.69 ± 0.78	5.21 ± 0.62 ^a^	3.44 ± 0.32 ^a^
O2		6.58 ± 1.04	6.04 ± 1.33 ^ab^	4.39 ± 1.27 ^ab^
O3		6.73 ± 1.77	7.73 ± 1.85 ^b^	5.42 ± 0.93 ^b^

^a,b^ Different superscripts in the same column mean significant difference at *p* < 0.05 level. O1: 106 μg OTA/kg feed, O2: 654 μg OTA/kg feed, O3: 1126 μg OTA/kg feed.

**Table 6 antioxidants-08-00178-t006:** Effect of Ochratoxin A treatment on glutathione peroxidase activity in liver and kidney homogenates (mean ± SD; *n* = 6).

**Liver (U/g 10,000 g Supernatant Protein)**				
	Day 0	Day 7	Day 14	Day 21
Control	1.76 ± 0.63	3.13 ± 1.17	3.15 ± 1.40	2.70 ± 0.99
O1		3.01 ± 0.96	3.05 ± 0.71	2.01 ± 0.71
O2		4.21 ± 1.59	3.91 ± 1.55	2.35 ± 1.27
O3		4.13 ± 1.45	4.76 ± 2.23	3.40 ± 1.04
**Kidney (U/g 10,000 g Supernatant Protein)**				
	Day 0	Day 7	Day 14	Day 21
Control	2.26 ± 0.43	2.24 ± 0.27 ^a^	2.79 ± 0.22	2.27 ± 0.71
O1		2.60 ± 0.26 ^ab^	2.69 ± 0.47	2.39 ± 0.11
O2		2.63 ± 0.28 ^ab^	2.59 ± 0.66	2.72 ± 0.28
O3		2.94 ± 0.26 ^b^	2.75 ± 0.48	2.62 ± 0.31

^a,b^ Different superscripts in the same column mean significant difference at *p* < 0.05 level. O1: 106 μg OTA/kg feed, O2: 654 μg OTA/kg feed, O3: 1126 μg OTA/kg feed.

**Table 7 antioxidants-08-00178-t007:** Effect of Ochratoxin A treatment on the relative expression of *KEAP1*, *NRF2* genes in the liver and kidney of broiler chickens (mean ± SD; *n* = 6 in a pool, equal amounts of cDNA per individual).

**Kelch-Like ECH-Associated Protein 1 (*KEAP1*)**				
	Day 0	Day 7	Day 14	Day 21
Liver				
Control	1.00 ± 0.05	0.71 ± 0.08	0.90 ± 0.08 ^a^	1.00 ± 0.16
O1		0.71 ± 0.10	0.93 ± 0.13 ^a^	0.91 ± 0.18
O2		0.69 ± 0.06	1.19 ± 0.15 ^b^	1.01 ± 0.14
O3		0.76 ± 0.14	1.27 ± 0.08 ^b^	1.09 ± 0.24
Kidney				
Control	1.00 ± 0.06	1.00 ± 0.14 ^b^	1.15 ± 0.09 ^b^	1.39 ± 0.22 ^b^
O1		0.97 ± 0.11 ^b^	0.95 ± 0.10 ^ab^	1.11 ± 0.20 ^a^
O2		0.73 ± 0.07 ^a^	0.92 ± 0.21 ^a^	0.94 ± 0.08 ^a^
O3		0.85 ± 0.05 ^ab^	0.96 ± 0.06 ^ab^	1.17 ± 0.13 ^ab^
**Nuclear Factor-Erythroid 2 p45-Related Factor 2 (*NRF2*)**				
	Day 0	Day 7	Day 14	Day 21
Liver				
Control	1.00 ± 0.06	1.19 ± 0.24 ^a^	1.52 ± 0.27 ^ab^	1.20 ± 0.20 ^a^
O1		1.07 ± 0.23 ^a^	1.21 ± 0.23 ^a^	1.21 ± 0.22 ^a^
O2		1.36 ± 0.28 ^ab^	1.23 ± 0.24 ^a^	1.46 ± 0.30 ^a^
O3		1.68 ± 0.25 ^b^	1.87 ± 0.22 ^b^	1.85 ± 0.31 ^b^
Kidney				
Control	1.01 ± 0.12	0.89 ± 0.07 ^a^	0.95 ± 0.19 ^a^	1.04 ± 0.12 ^a^
O1		0.79 ± 0.14 ^a^	0.92 ± 0.15 ^a^	0.95 ± 0.16 ^a^
O2		0.80 ± 0.13 ^a^	0.98 ± 0.15 ^a^	0.96 ± 0.09 ^a^
O3		1.22 ± 0.14 ^b^	1.39 ± 0.25 ^b^	1.29 ± 0.23 ^b^

^a,b^ Different superscripts in the same column mean significant difference at *p* < 0.05 level. O1: 106 μg OTA/kg feed, O2: 654 μg OTA/kg feed, O3: 1126 μg OTA/kg feed.

**Table 8 antioxidants-08-00178-t008:** Effect of Ochratoxin A treatment on the relative expression of *GPX3*, *GPX4* genes in liver and kidney of broiler chickens (mean ± SD; *n* = 6 in a pool, equal amounts of cDNA per individual).

**Glutathione Peroxidase 3 (*GPX3*)**				
	Day 0	Day 7	Day 14	Day 21
Liver				
Control	1.00 ± 0.03	0.84 ± 0.10 ^b^	0.99 ± 0.16 ^a^	1.49 ± 0.14 ^c^
O1		0.59 ± 0.15 ^a^	0.95 ± 0.18 ^a^	0.86 ± 0.10 ^a^
O2		0.73 ± 0.11 ^ab^	1.05 ± 0.11 ^ab^	0.89 ± 0.12 ^a^
O3		0.84 ± 0.19 ^b^	1.24 ± 0.09 ^a^	1.24 ± 0.13 ^b^
Kidney				
Control	1.00 ± 0.03	0.83 ± 0.06 ^a^	0.94 ± 0.08 ^a^	1.05 ± 0.06 ^a^
O1		0.90 ± 0.08 ^a^	0.99 ± 0.01 ^ab^	1.10 ± 0.11 ^a^
O2		0.96 ± 0.11 ^a^	0.99 ± 0.09 ^ab^	1.00 ± 0.08 ^a^
O3		1.60 ± 0.10 ^b^	1.56 ± 0.14 ^b^	1.47 ± 0.15 ^b^
**Glutathione Peroxidase 4 (*GPX4*)**				
	Day 0	Day 7	Day 14	Day 21
Liver				
Control	1.00 ± 0.05	0.51 ± 0.02	0.40 ± 0.03 ^ab^	0.53 ± 0.03
O1		0.49 ± 0.02	0.32 ± 0.01 ^a^	0.51 ± 0.03
O2		0.50 ± 0.05	0.42 ± 0.03 ^b^	0.54 ± 0.04
O3		0.51 ± 0.06	0.42 ± 0.10 ^ab^	0.49 ± 0.01
Kidney				
Control	1.00 ± 0.05	0.48 ± 0.03 ^b^	0.52 ± 0.03 ^c^	1.25 ± 0.04 ^c^
O1		0.49 ± 0.02 ^b^	0.32 ± 0.01 ^a^	0.90 ± 0.03 ^b^
O2		0.40 ± 0.02 ^a^	0.35 ± 0.02 ^ab^	0.72 ± 0.04 ^a^
O3		0.37 ± 0.03 ^a^	0.36 ± 0.03 ^b^	0.75 ± 0.04 ^a^

^a,b^ Different superscripts in the same column mean significant difference at *p* < 0.05 level. O1: 106 μg OTA/kg feed, O2: 654 μg OTA/kg feed, O3: 1126 μg OTA/kg feed.

**Table 9 antioxidants-08-00178-t009:** Effect of Ochratoxin A treatment on the relative expression of *GSS* and *GSR* genes in the liver and kidney of broiler chickens (mean ± SD; *n* = 6 in a pool, equal amounts of cDNA per individual).

**Glutathione Synthetase (GSS)**				
	Day 0	Day 7	Day 14	Day 21
Liver				
Control	1.00 ± 0.03	0.73 ± 0.09	1.25 ± 0.17 ^b^	0.93 ± 0.13
O1		0.71 ± 0.09	0.83 ± 0.09 ^a^	0.76 ± 0.31
O2		0.74 ± 0.10	1.06 ± 0.19 ^ab^	0.78 ± 0.20
O3		0.70 ± 0.12	1.19 ± 0.25 ^b^	0.57 ± 0.10
Kidney				
Control	1.00 ± 0.08	0.96 ± 0.14 ^ab^	1.11 ± 0.07 ^b^	0.84 ± 0.17
O1		1.08 ± 0.17 ^b^	0.83 ± 0.09 ^a^	0.79 ± 0.14
O2		0.81 ± 0.15 ^a^	1.01 ± 0.10 ^b^	0.82 ± 0.08
O3		0.82 ± 0.16 ^a^	0.98 ± 0.13 ^ab^	0.80 ± 0.07
**Glutathione Reductase (GSR)**				
	Day 0	Day 7	Day 14	Day 21
Liver				
Control	1.00 ± 0.10	0.48 ± 0.04	0.60 ± 0.06 ^b^	0.47 ± 0.05
O1		0.51 ± 0.09	0.34 ± 0.06 ^a^	0.52 ± 0.06
O2		0.54 ± 0.04	0.60 ± 0.09 ^b^	0.48 ± 0.03
O3		0.50 ± 0.08	0.54 ± 0.05 ^b^	0.47 ± 0.08
Kidney				
Control	1.00 ± 0.04	0.99 ± 0.10	0.95 ± 0.13 ^b^	1.10 ± 0.16
O1		1.06 ± 0.21	0.75 ± 0.12 ^a^	1.04 ± 0.16
O2		0.94 ± 0.12	0.93 ± 0.04 ^b^	0.99 ± 0.16
O3		1.11 ± 0.19	0.91 ± 0.09 ^b^	1.04 ± 0.20

^a,b^ Different superscripts in the same column mean significant difference at *p* < 0.05 level. O1: 106 μg OTA/kg feed, O2: 654 μg OTA/kg feed, O3: 1126 μg OTA/kg feed.

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
