# Peer review of "Long-Term Effects of Ochratoxin A on the Glutathione Redox System and Its Regulation in Chicken"

_antioxidants, 2019, doi:10.3390/antiox8060178_

Round 1
Reviewer 1 Report
In this paper. authors conducted a well organized study both in terms of methods and statistical analysis. The introduction does provide sufficient background and include all relevant references. The methods are adequately described, the results clearly presentedand the conclusions are supported by the results. However, it doesn't provide a real innovative insight to the topic.
Author Response
Thank you for the comments. The manuscript re-checked again for spelling and grammar mistakes.
The innovative insight of the topic was not a purpose of this study, but we try to add some proposals at the end of the discussion (r. 335-338), which probably useful for further studies and possibly would be innovative.
Reviewer 2 Report
The manuscript by Kovesi et al described a study aiming to evaluate the effect of ochratoxin A exposure on the in vivo reactive oxidative stress level, and its influcence on several redox sensitive protein gene transcription level regulations. The authors found that OTA initiates free radical formation, and severity depends on the exposure dose and time. GSH reduced form cocentration increased in response to stimulation of increased ROS by OTA. In addition, GPX protein expression level was also upregulated, but not in blood only in liver and kidney.
Can the author explain the mechanism why GPx activity didn't change in blood but in liver and kidney?
Author Response
Thank you for your comments about our paper.
The answer to your question: "Can the author explain the mechanism why GPx activity didn't change in blood but in liver and kidney?" you can find in the revised manuscript on r. 320-323 as possible explanation.